Corrected: Author correction

# Integrated molecular subtyping defines a curable oligometastatic state in colorectal liver metastasis

Sean P. Pitroda[1,2], Nikolai N. Khodarev[1,2], Lei Huang[3], Abhineet Uppal[4], Sean C. Wightman[4], Sabha Ganai[5], Nora Joseph[6], Jason Pitt[7], Miguel Brown[7], Martin Forde[7], Kathy Mangold [6], Lai Xue[4], Christopher Weber[8], Jeremy P. Segal[8], Sabah Kadri[8], Melinda E. Stack[4], Sajid Khan[9], Philip Paty[10], Karen Kaul[6], Jorge Andrade[3], Kevin P. White[7,11], Mark Talamonti[12], Mitchell C. Posner[4], Samuel Hellman[1,2] & Ralph R. Weichselbaum[1,2]

The oligometastasis hypothesis suggests a spectrum of metastatic virulence where some metastases are limited in extent and curable with focal therapies. A subset of patients with metastatic colorectal cancer achieves prolonged survival after resection of liver metastases consistent with oligometastasis. Here we define three robust subtypes of de novo colorectal liver metastasis through integrative molecular analysis. Patients with metastases exhibiting MSI-independent immune activation experience the most favorable survival. Subtypes with adverse outcomes demonstrate VEGFA amplification in concert with (i) stromal, mesenchymal, and angiogenic signatures, or (ii) exclusive NOTCH1 and PIK3C2B mutations with E2F/MYC activation. Molecular subtypes complement clinical risk stratification to distinguish low-risk, intermediate-risk, and high-risk patients with 10-year overall survivals of 94%, 45%, and 19%, respectively. Our findings provide a framework for integrated classification and treatment of metastasis and support the biological basis of curable oligometastatic colorectal cancer. These concepts may be applicable to many patients with metastatic cancer.

[1] Department of Radiation and Cellular Oncology, The University of Chicago, Chicago, IL 60637, USA. [2] Ludwig Center for Metastasis Research, The University of Chicago, Chicago, IL 60637, USA. [3] Center for Research Informatics, The University of Chicago, Chicago, IL 60637, USA. [4] Department of Surgery, The University of Chicago, Chicago, IL 60637, USA. [5] Department of Surgery, Southern Illinois University, Springfield, IL 62702, USA. [6] Department of Pathology, NorthShore University Hospital, Evanston, IL 60201, USA. [7] Institute for Genomics and Systems Biology, The University of Chicago, Chicago, IL 60637, USA. [8] Department of Pathology, The University of Chicago, Chicago, IL 60637, USA. [9] Department of Surgery, Yale School of Medicine, New Haven, CT 06510, USA. [10] Department of Surgery, Memorial Sloan-Kettering Cancer Center, New York, NY 10065, USA. [11] Tempus Labs, Chicago, IL 60654, USA. [12] Department of Surgery, NorthShore University Hospital, Evanston, IL 60201, USA. These authors contributed equally: Sean P. Pitroda, Nikolai N. Khodarev. Correspondence and requests for materials should be addressed to R.R.W. (email: rrw@radonc.uchicago.edu)

Metastases are the leading cause of cancer-related deaths and frequently are widely disseminated, which has led to the prevailing view that metastases are always widespread. The oligometastasis hypothesis suggests that metastatic spread is a spectrum of virulence where some metastases are limited both in number and organ involvement and potentially curable with surgical resection or other loco-regional therapies[1,2]. This paradigm is in stark contrast to the outcomes of patients with solid tumors where widespread metastases are largely fatal despite recent advances in systemic therapy. To date, the oligometastasis concept has been challenged, in large part, due to the lack of supporting molecular data to identify metastases associated with restricted spread[3,4].

Limited metastasis is relatively common. Data from clinical trials and single institution analyses of lung, breast, colorectal, prostate and renal cancers suggest that as many as 40–60% of patients with metastasis present with or develop limited disease[5–8]. Patients with limited liver metastases from colorectal cancer (CRC) have been consistently demonstrated to achieve prolonged survival after hepatic resection[9,10] and provide an opportunity to investigate the molecular basis for oligometastasis. While there have been extensive investigations into the molecular subtypes of primary human cancers, little is known regarding molecular subtypes of metastasis and their relation to clinical outcomes. Here, utilizing independent clinical cohorts of CRC patients who underwent resection of liver metastases, we identified integrated molecular patterns in liver metastases associated with long-term survival. Our findings indicate a molecular basis for oligometastasis that is predictive of clinical outcome and complementary to established clinical risk factors associated with long-term survival following hepatic resection. Our findings may have important clinical implications in the selection of local therapy for those patients with potentially curable oligometastatic disease from those whose few metastases are a part of a large cascade of widespread disease. These concepts may be applicable to many histological types of cancer.

## Results

**Clinical characteristics and patient outcomes.** One hundred thirty-four patients with comprehensive clinical annotations underwent hepatic resection of limited CRC liver metastases (CRCLM). The clinical characteristics of these patients are summarized in Table 1. The median patient age was 61 years (range, 29–85). Patients were diagnosed with primary adenocarcinoma of the colon (72%) or rectum (28%) and presented with either synchronous (47%) or metachronous (53%) liver metastasis. The initial number of liver metastases was one in 61%, two in 22% and three or more in 17% of patients. Liver metastases were limited to one hepatic lobe in 91% of patients and two hepatic lobes in 9% of patients. Our analysis focused on de novo liver metastases and excluded patients with extrahepatic disease or a history of previously resected metastasis. Patients received uniform treatment with 5-fluorouracil-based perioperative chemotherapy, curative intent management of primary colorectal tumors, and partial hepatectomy of all visible liver metastases (Table 1). Postoperatively all patients were surveilled with serial axial CT imaging and serum CEA levels.

At a median follow-up of 49 months, 32% of patients had no evidence of metastatic recurrence. These patients had a 10-year OS of 77% whereas patients with clinically evident, recurrent metastases exhibited a 10-year OS of 13% ($P < 0.0001$, log-rank test) (Fig. 1a). We calculated Clinical Risk Scores (CRS), a widely utilized prognostic tool for CRC patients undergoing liver metastasis resection[9,11,12], using the following adverse clinical and pathological features: (1) disease-free interval between

primary tumor diagnosis and development of metastasis <12 months, (2) number of liver metastases >1, (3) largest liver metastasis >5.0 cm, (4) lymph node-positive primary CRC, and (5) CEA >200 ng mL$^{-1}$. 34% of patients exhibited a low CRS (less than two adverse features). As expected, OS was significantly greater for patients with low versus high CRS (two or more adverse features) (10-year: 62% vs. 22%, $P = 0.0008$, log-rank test) (Fig. 1b). These outcomes were consistent with those previously reported in the literature[9]. In this context, we investigated whether the intrinsic molecular features of CRCLM enhance the identification of patients with long-term survival after hepatic resection of limited metastases.

### Table 1 Clinical and pathological characteristics of colorectal cancer patients

| Clinicopathological variable | Clinical cohort (n = 134) |
| --- | --- |
| Age (median, range) | 61 (29–85) |
| Sex | |
| Male | 57% |
| Female | 43% |
| Primary tumor | |
| Colon | 72% |
| Rectum | 28% |
| Metastatic presentation | |
| Synchronous | 47% |
| Metachronous | 53% |
| Tumor size | |
| ≤5 cm | 78% |
| >5 cm | 22% |
| Primary lymph node status | |
| Negative | 36% |
| Positive | 64% |
| Initial number of liver metastases | |
| 1 | 61% |
| 2 | 22% |
| 3+ | 17% |
| Disease-free interval from primary tumor to metastasis | |
| <12 mo | 61% |
| ≥12 mo | 39% |
| CEA level | |
| <200 ng/mL | 95% |
| ≥200 ng/mL | 5% |
| Clinical risk scores (CRS) | |
| <2 | 34% |
| ≥2 | 66% |
| Hepatic involvement | |
| Unilobar | 91% |
| Bilobar | 9% |
| Extent of resection | |
| Wedge/segmentectomy | 58% |
| Lobectomy/extended lobectomy | 42% |
| Resection margin | |
| Negative | 85% |
| Positive | 15% |
| Peri-operative chemotherapy | 98% |
| Follow-up (mo) (median, range) | 49 (4.3–328) |
| Metastatic recurrence | 68% |
| Patterns of failure | |
| Liver only | 38% |
| Liver and lung | 34% |
| Other sites (e.g., peritoneum, bone, adrenal, brain) | 28% |
| Death event | 58% |

Clinical and pathological characteristics of patients with liver metastases from colorectal cancer selected for study

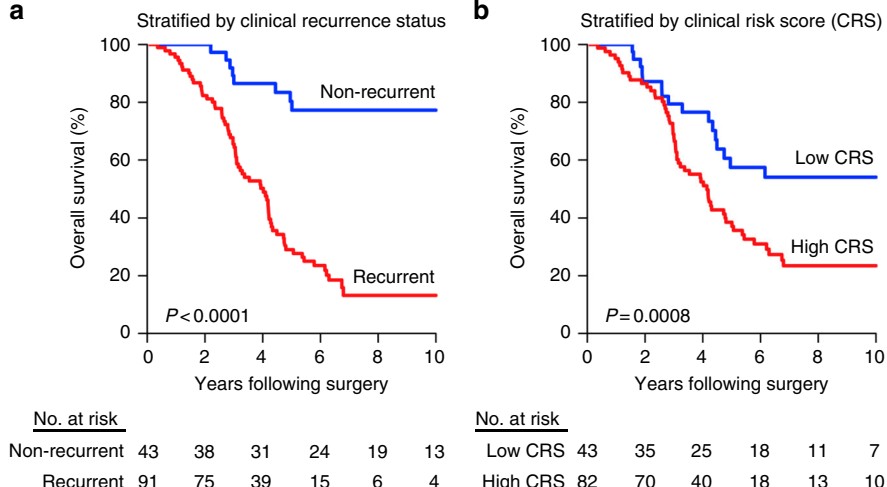

**Fig. 1** Clinical outcomes following surgical resection of limited liver metastases from colorectal cancer. Kaplan–Meier curves of overall survival by **a** clinical recurrence status (as determined by post-operative surveillance CT imaging and serum CEA measurements) or **b** clinical risk scores (CRS) following hepatic resection of limited de novo CRCLM. Low CRS was defined as values less than two. *P*-values were determined using log-rank tests

**Consensus molecular subtypes of primary colorectal cancer.** Gene expression analysis is an established approach for molecular subtyping of primary human cancers[13,14]. The International Colorectal Cancer Subtyping Consortium (CRCSC) demonstrated the existence of four biologically and clinically distinct consensus molecular subtypes (CMS) of CRC based on gene expression analysis of 3962 primary tumors[15,16]. However, it is unknown whether CMS subtypes also exist in CRCLM. We first validated the application of CMS classification to the analysis of RNA Sequencing data from 558 primary CRC tumors in The Cancer Genome Atlas (TCGA)[17], which verified the expected frequencies of CMS subtypes (Fig. 2a). We then applied the single-sample CMS classifier to the analysis of two independent CRCLM datasets derived from RNA Sequencing analysis of 93 patients in our study (Cohorts UC and NS in Fig. 2a), which demonstrated CMS2 and CMS4 patterns in 62 and 12% of liver metastases with a notable absence of CMS1 (1%) and CMS3 (0%) patterns (Fig. 2a). We examined whether this result was related to selection bias for patients with limited, resectable metastatic disease or was generalizable to widely metastatic or unresectable CRCLM. In five independent datasets comprising an additional 234 CRCLMs derived from either hepatic resection (60% of samples) or biopsy due to unresectable or widely metastatic disease (40% of samples), we observed similar frequencies of CMS2 and CMS4 patterns in 60 and 7% (vs. 37 and 23% of primary CRC). CMS1 and CMS3 subtypes comprised 2 and 1% of liver metastases (vs. 14 and 13% of primary CRC). In addition, 30% of CRCLM were unclassifiable based on CMS subtypes (Fig. 2a). While CMS classes were associated with distinct clinical outcomes in primary CRCs, we observed no association between CMS subtypes and OS in patients with resected liver metastases (Supplementary Fig. 2).

**Integrated subtypes of colorectal liver metastasis.** Transcriptomic analyses using consensus clustering of individual mRNA or miRNA datasets were limited in the molecular subtyping of colorectal liver metastases (Supplementary Figs 3 and 4). Based on previous work which demonstrated coordinated miRNA-mRNA transcriptional networks underlying metastatic phenotypes, as well as primary CRC subtypes[18–21], we performed an integrated expression analysis to uncover novel intrinsic subtypes of CRCLM. We utilized the similarity network fusion (SNF) algorithm to incorporate parallel miRNA and mRNA networks in 93 patient samples independently of clinical,

pathological, or survival data. SNF is a computational method for integration of diverse types of data with superior performance in the identification of cancer subtypes when compared to single data and established integrative approaches[22]. We identified three distinct molecular subtypes of CRCLM denoted SNF1 (subtype 1; 33%), SNF2 (subtype 2; 28%), and SNF3 (subtype 3; 39%) (Fig. 2b). Despite the detection of subtypes solely based on molecular features, we found the SNF subtypes exhibited heterogeneous clinical outcomes with 10-year OS of 37%, 64%, and 20%, respectively (*P* = 0.021, log-rank test) (Fig. 2c and Supplementary Fig. 6). Using permutation analysis, we examined the importance of the SNF cluster structure on its association with OS. After 1000 randomized mRNA-miRNA permutations, we found the SNF clustering algorithm was unlikely to generate our empirical OS difference by chance (empirical *P* = 0.0007) (Supplementary Fig. 7). Importantly, patients with subtype 2 metastases experienced fewer metastatic recurrences or deaths after hepatic resection as compared to subtype 1 or subtype 3 metastases (Fig. 2d and Supplementary Fig. 8), and metastatic recurrences of subtype 2 metastases were significantly more likely to be limited in number, defined as 1–3 subsequent metastases, as compared to subtype 1 or subtype 3 metastases (Fig. 2d).

Each subtype demonstrated distinct patterns of mRNA and miRNA expressions (Fig. 2e). Subtype 2 and subtype 3 metastases displayed similar patterns of primary CRC CMS subtypes whereas, by contrast, subtype 1 metastases almost exclusively exhibited the CMS2 pattern (Supplementary Fig. 9). However, only 10, 5.6, and 16% of subtypes 1-specific, 2-specific, and 3-specific gene sets overlapped with the CMS classifier suggesting that molecular subtyping provided unique classification of CRC tumors. In addition, the type of perioperative chemotherapy had no effect on the molecular patterns (Supplementary Fig. 10) or overall survival of resected CRCLM. Given that independent datasets of integrated molecular data do not exist for clinically annotated CRCLM, we trained and validated an mRNA-based classifier to identify subtype 2-specific transcriptional patterns in patient samples. We found the molecular classifier accurately identified subtype 2 patients with 100% sensitivity and 81% specificity in our patient cohort (Supplementary Fig. 11A,B). In a separate dataset of CRCLM patients with similar clinical and pathological features treated with hepatic resection at the Memorial Sloan-Kettering Cancer Center (*n* = 96), metastases classified as subtype 2 were confirmed to demonstrate favorable

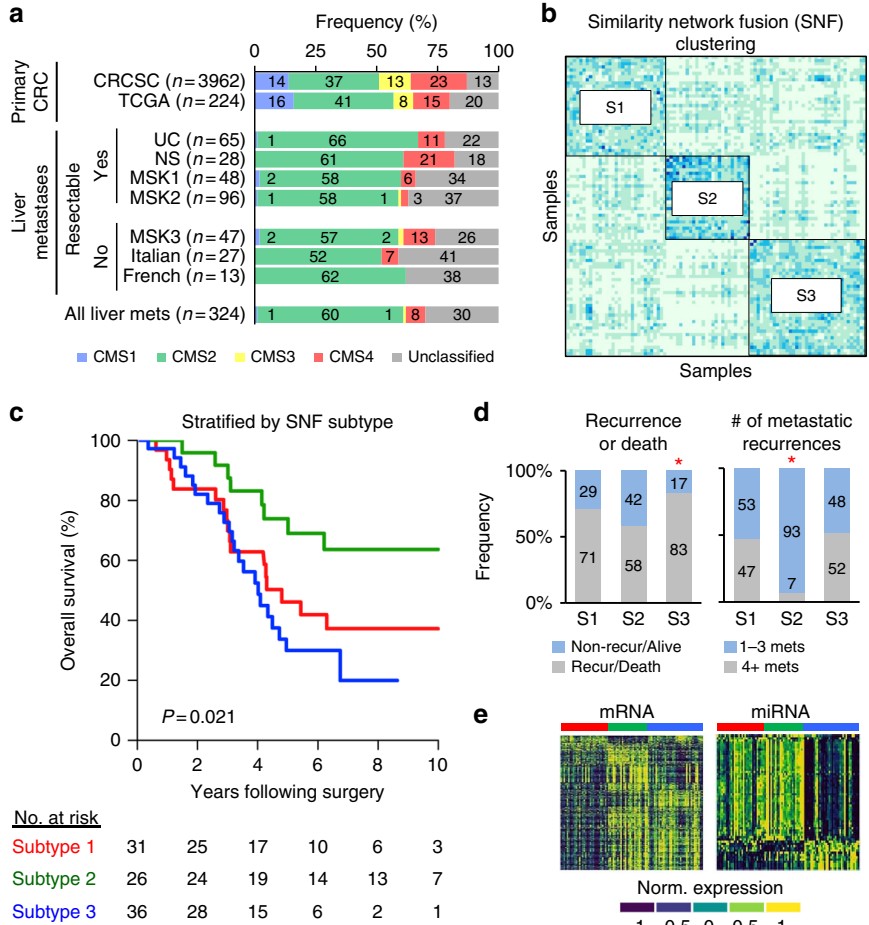

**Fig. 2** Identification of intrinsic molecular subtypes of colorectal liver metastases. **a** Consensus molecular subtypes (CMS) of primary colorectal cancers obtained from the Colorectal Cancer Subtyping Consortium (CRCSC) or calculated in primary colorectal cancers of The Cancer Genome Atlas (TCGA). CMS subtypes were also determined in colorectal liver metastases from patients undergoing partial hepatectomy of resectable liver metastases (UC, NS, MSK1, and MSK2 cohorts) or biopsy of unresectable liver metastases (MSK3, Italian and French cohorts). Cohorts contain independent clinical and molecular datasets (see Supplementary Information). **b** Consensus clustering based on similarity network fusion (SNF) subtyping of colorectal liver metastases. Subtype 1 = S1, Subtype 2 = S2, Subtype 3 = S3. **c** Kaplan–Meier curves of overall survival by molecular subtype. *P*-value was determined using a log-rank test. **d** Metastatic recurrence patterns by molecular subtype. Asterisks denote statistical significance based on Fisher's exact test for each subtype versus the two other subtypes. **e** Differentially expressed mRNAs (left) and miRNAs (right) between the three molecular subtypes (see Supplementary Data 2 and 3). Subtype 1 = red, Subtype 2 = green, Subtype 3 = blue

clinical outcomes as compared to metastases with subtypes 1 or 3 patterns (Supplementary Fig. 11C). These findings supported integrated molecular subtypes and their associations with clinical outcomes in an independent dataset from a distinct institution.

**Molecular characterization of liver metastasis subtypes.** Ensemble of Gene Set Enrichment Analyses (EGSEA) provided substantial insight into the biological features of the subtypes of CRC liver metastases (Fig. 3a). EGSEA quantifies the enrichment of biologically defined gene sets within a gene expression profile[23]. We found subtype 3 metastases showed enrichment for expression patterns associated with high stromal infiltration, epithelial-mesenchymal transition (EMT), extracellular matrix remodeling, angiogenesis, inflammatory response, and KRAS signaling (Fig. 3a). Subtype 2 metastases similarly exhibited enrichment for EMT and KRAS pathways; however, these metastases were distinguished by high immune infiltration, enrichment of interferon alpha and gamma signatures, and activation of p53 pathways. In concert with these findings, subtype 2 significantly overexpressed innate and adaptive immune genes,

such as those which mediate T cell activation and crosstalk between antigen presenting cells and T cells, as compared to subtype 1 and subtype 3 (Fig. 3b and Supplementary Data 6). Subtype 1 metastases displayed both low stromal and low immune infiltration signatures but were markedly enriched for E2F/MYC signaling, including *TERT* (telomerase) overexpression, as well as abnormalities in DNA damage signaling and cell cycle checkpoints.

Importantly, CRCLM subtypes were also discernible at the histological level (Supplementary Fig. 12). Subtype 2 metastases exhibited dense band-like peritumoral infiltration of CD3-positive and CD8-positive lymphocytes extending intratumorally, and trichrome staining demonstrated minimal fibrosis, whereas subtype 3 metastases were distinguished by marked intratumoral and peritumoral fibrosis which harbored peritumorally restricted lymphocytic infiltrate. In contrast, subtype 1 metastases revealed prominent nests of tumor cells with minimal CD3-positive or CD8-positive cells or fibrosis. These findings demonstrated unique properties of molecular subtypes associate with differential patient outcomes after hepatic resection of limited CRCLM.

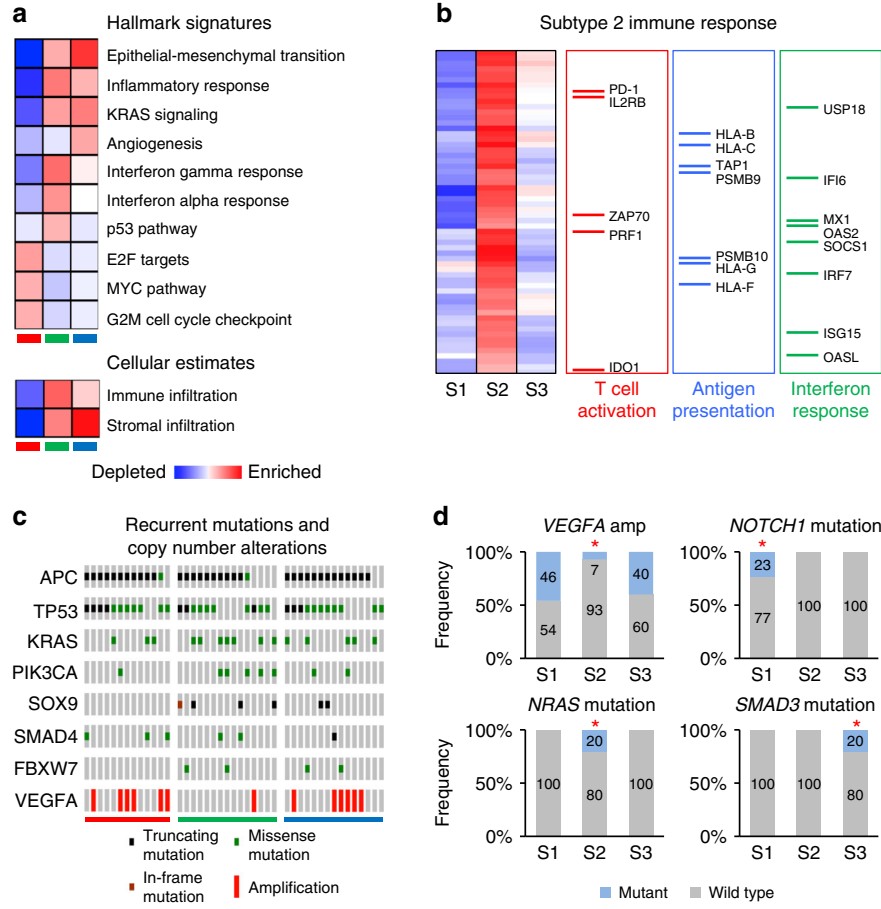

**Fig. 3** Molecular signatures of intrinsic subtypes of colorectal liver metastases. **a** Ensemble of gene set enrichment analyses (EGSEA) of significantly enriched "Hallmark" and "Cellular Estimate" gene signatures within each molecular subtype. Red color denotes enrichment, while blue color indicates depletion. Color intensity in EGSEA heatmaps is proportional to significance level (see Supplementary Data 4 and 5). Subtype 1 = red, Subtype 2 = green, Subtype 3 = blue. **b** Functional categorization of differentially expressed immune genes overexpressed in subtype 2. In heatmap red color denotes overexpression, while blue color indicates suppression. **c** OncoPrint plot of recurrent colorectal cancer mutations and copy number alterations by subtype. **d** Frequencies of subtype-specific genomic alterations. Asterisks denote statistical significance based on Fisher's exact test comparing each subtype group to the two other subtypes. Subtype 1 = S1, Subtype 2 = S2, Subtype 3 = S3

**Mutational and copy number landscapes.** Fifty-nine liver metastases and matched normal liver specimens underwent next-generation genomic sequencing using OncoPlus, a clinically validated hybrid capture genomic sequencing platform comprising 1212 commonly altered cancer genes for mutational and copy number analyses[24]. Mutation significance (MutSig) analysis confirmed enrichment in CRC driver gene mutations of *APC*, *TP53*, *KRAS*, *PIK3CA*, *SOX9*, *SMAD4*, and *FBXW7* in 83%, 73%, 37%, 20%, 14%, 14% and 12% of liver metastases, respectively (Fig. 3c, Supplementary Fig. 13, and Supplementary Data 7). In addition, we observed frequent gene-level copy number variations, including amplifications of *VEGFA* (Fig. 3c), *MYC*, and *ERBB2* and deletion of *MAP2K4*, which were previously described for primary CRC[17]. We also found mutational patterns reflected the anatomic location of the primary tumor origin of liver metastases. Eighty-three percent of liver metastases from right-sided colon cancers exhibited activating somatic mutations in *KRAS*, in contrast to 24% from left-sided cancers (*P* = 0.0005, Fisher's exact test). Also, *PIK3CA* mutations occurred in 50% of metastases derived from right-sided primary tumors versus 8% from left-sided primary tumors (*P* = 0.0038, Fisher's exact test).

We extended these findings by characterizing the mutational and copy number landscapes of CRCLM by molecular subtypes. We identified unique somatic mutations in each subtype (Fig. 3d

and Supplementary Data 8). Subtype 3 demonstrated exclusive somatic mutations in *SMAD3*, whereas *NOTCH1* and *PIK3C2B* mutations occurred only in subtype 1. By contrast, *NRAS*, *CDK12*, and *EBF1* mutations were unique to subtype 2 (Fig. 3d). In addition, amplification of *VEGFA* was more prevalent in subtypes 1 and 3 as compared to subtype 2 (Fig. 3d and Supplementary Data 8). Notably, we found no significant differences in the frequency of *KRAS* or *BRAF* mutations across subtypes. Taken together, these data support the notion that subtypes of CRCLM harbor distinct genomic aberrations.

Furthermore, we found the median number of mutations per sample was not statistically different across molecular subtypes. Given that mismatch repair deficiency leading to microsatellite instability (MSI) contributes to tumor hypermutation in association with cytotoxic immune infiltration[25], we investigated whether MSI explained the immune subtype 2. We identified an MSI phenotype in 3.4% of patients, which is consistent with the incidence of MSI in metastatic colorectal cancer[26]. However, only one subtype 2 metastasis demonstrated an MSI-high phenotype, while two metastases—one from subtype 1 and one from subtype 2, exhibited an MSI-low phenotype. The subtype 2 MSI-high and MSI-low metastases, but not subtype 1 MSI-low metastasis, showed significant enrichment of cytotoxic cell signature expression (Supplementary Fig. 14). Although mutational burden did not

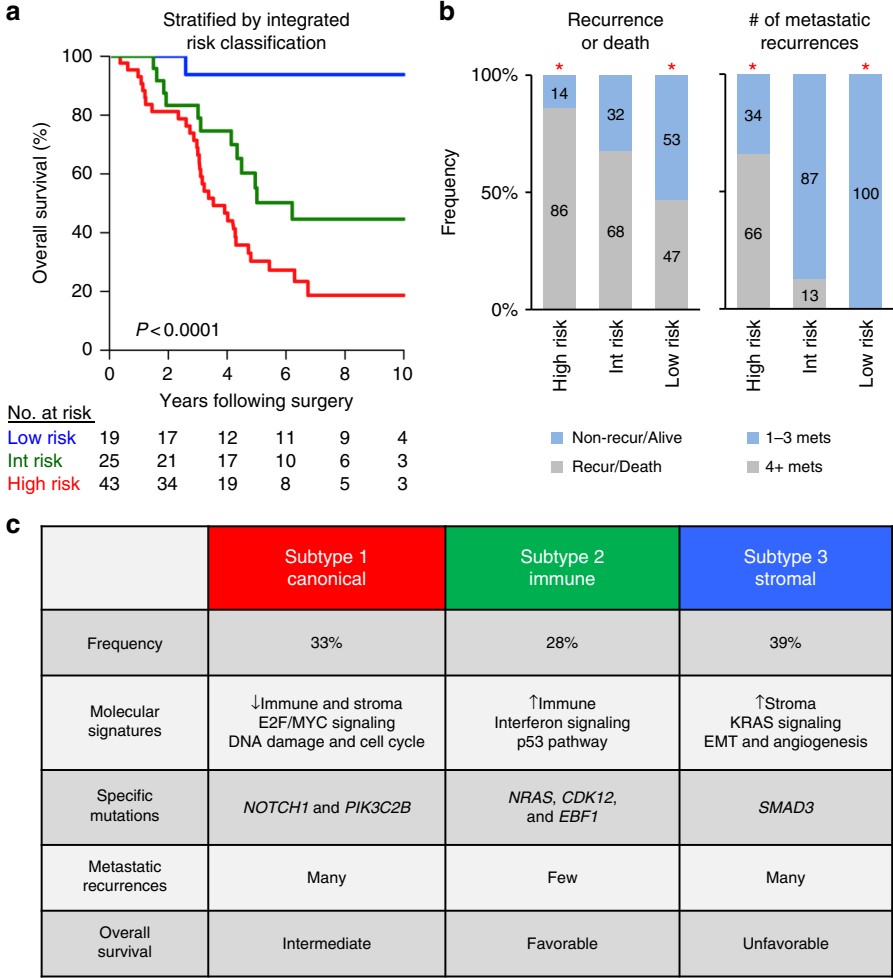

**Fig. 4** Integration of intrinsic molecular subtypes and clinical risk stratification. **a** Kaplan–Meier curves of overall survival following initial hepatic resection of limited de novo CRCLM based on integrated risk classification of molecular subtype and clinical risk scores (CRS). *P*-value was determined using a log-rank test. **b** Metastatic recurrence patterns for integrated risk groups. Asterisks denote statistical significance based on Fisher's exact test for each individual group versus the two additional groups. **c** Proposed classification of colorectal liver metastasis based on integrated molecular subtypes

correspond to the subtype 2 immune subtype, increased mutational burden associated with MSI in subtype 2, including hypermutation (*n* = 1760 mutations) in the MSI-low metastasis related to *POLE* mutation. In addition, recent data suggest that specific mutations also predict cytotoxic immune responses[27]. In this regard, *ARID2* mutations are associated with increased cytolytic activity in multiple types of cancers. In our dataset, we also found a significant elevation in cytotoxic immune responses in patients with somatic *ARID2* mutations (Supplementary Fig. 14). As well, metastases with subtype 2-specific mutations in *CDK12*, *NRAS*, or *EBF1*, in contrast to metastases with subtype 1- or subtype 3-specific mutations in *NOTCH1*, *PIK3C2B*, or *SMAD3*, showed significant enrichment for cytotoxic immune responses (*P* = 0.034, Student's *t*-test). Taken together, these findings identify novel mutations associated with immune responses and favorable clinical outcomes of CRCLM.

**Integrated risk stratification**. We investigated whether molecular subtyping could improve clinical risk stratification after hepatic resection of CRCLM by augmenting the prognostic effect of CRS. Multivariate Cox proportional hazard analysis indicated the prognostic impact of SNF-based molecular subtypes was statistically complementary to CRS (Supplementary Fig. 15). Integration of molecular subtypes and CRS yielded three prognostic risk groups: (1) low-risk (22% of patients)—subtypes 1 and 2 with low

CRS; (2) intermediate-risk (29% of patients)—subtype 2 with high CRS and subtype 3 with low CRS; (3) high-risk patients (49% of patients)—subtypes 1 and 3 with high CRS (Fig. 4a and Supplementary Fig. 15). Multivariate Cox proportional hazard analyses incorporating primary tumor anatomic site, type of perioperative chemotherapy, treatment year, or mutational data, including *KRAS* mutation, *BRAF* mutation, or MSI, did not independently contribute to prognostication in our cohort. Ten-year OS for low-risk, intermediate-risk, and high-risk groups were 94%, 45%, and 19%, respectively, at median follow-up times of 76, 54, and 40 months (Fig. 4a). Notably, while patients with subtype 1 metastases generally demonstrated unfavorable clinical outcomes, a subset of subtype 1 metastases with low CRS (23%) achieved long-term survival, which correlated with a reduced frequency of extrahepatic metastatic recurrence in these patients in contrast to subtype 1 metastases with high CRS (33% vs. 81%). While distant metastasis-free survival significantly differed across risk groups (median value—low-risk: 59 mo. vs. intermediate-risk: 35 mo. vs. high-risk: 13 mo; *P* = 0.0021, log-rank test) (Supplementary Fig. 16A), 47% of low-risk, 68% of intermediate-risk, and 86% of high-risk patients developed subsequent meta-static recurrence after hepatic resection (Supplementary Fig. 16B). Importantly, metastatic recurrences were limited in number in 100% of low-risk patients in contrast to 87% of intermediate-risk and 34% of high-risk patients (Fig. 4B) (*P* < 0.0001, Chi-square

test across groups). These findings demonstrate molecular subtypes of CRCLM significantly improve clinical risk stratification for the identification of patients with favorable prognoses after hepatic resection of limited de novo CRCLM.

## Discussion

We performed a multilevel, genome-wide investigation into the molecular basis of liver metastases from primary CRC. We validated the presence of primary CRC expression subtypes, mutations, and gene-level copy number variations within CRCLM, but found these molecular features to have limited prognostic utility in metastatic outcomes. Our findings support the notion that the biological features important for primary tumor growth differ from those enabling successful metastatic dissemination. In contrast, integrated transcriptional analysis of mRNA and miRNA networks identified three distinct subtypes of CRCLM which complement clinical risk stratification for prediction of long-term survival following hepatic resection of limited liver metastases. This predictive power is based on the identification of intrinsic molecular subtypes of de novo CRCLM, which associate with distinct genomic patterns, metastatic recurrences and clinical outcomes. Based on these findings, we propose the first classification of clinical metastasis for CRCLM based on SNF subtypes and their associated molecular and clinical features: Subtype 1 = Canonical; Subtype 2 = Immune; Subtype 3 = Stromal (Fig. 4c).

The concept of limited, curable metastasis has not always been accepted due, in part, to the paucity of molecular evidence to identify this unique entity. Our analysis focused on de novo liver metastases as the initial site of metastasis in patients with no prior history of metastatic disease. In this subgroup of patients, cure was achieved in ~30% of patients following hepatic resection, which is consistent with published studies[9]. Previous work has compared the molecular patterns of matched primary CRC and liver metastases or consecutive metastases within an individual patient[28,29]. In contrast, we characterized inter-metastatic heterogeneity across individual patients to identify intrinsic molecular determinants correlated with clinical outcomes. Importantly, our findings uncover previously unrecognized molecular subtypes of clinical metastasis and support the notion that the molecular heterogeneity of colorectal liver metastases contributes to differences in clinical outcomes for patients.

The molecular subtypes of CRCLM emerged from our analysis (Fig. 4c). We identified a relatively indolent immune-enriched subtype of CRCLM, denoted subtype 2, which developed clinically evident metastases limited in number. We also determined MSI-independent mutations in subtype 2 metastases which associate with increased cytotoxic immune responses. Subtype 3, a stroma-enriched subtype of CRCLM, displayed well-characterized pro-metastatic pathways mediating EMT and angiogenesis in association with poor clinical outcomes. A canonical molecular subtype associated with E2F/MYC, DNA damage-related and cell cycle signaling showed variable metastatic recurrence patterns and clinical outcomes. Integrated risk classification by subtype and clinical risk scores (CRS) revealed a low-risk group, defined by subtype 1 or 2 molecular patterns in association with low CRS, with a 10-year overall survival of ~95% (Fig. 4a). This integrated low-risk group is most consistent with an oligometastatic phenotype.

Based on our findings, it might be speculated that there are intrinsic tumor and extrinsic host factors that determine metastatic virulence and that our indolent immune-enriched subtype represents a form of immunological equilibrium[30,31]. These immunologic features are contrasted with subtypes 1 and 3, which do not exhibit immune markers and, thus, have escaped due, in part, to failed immuno-editing, as well as tumor intrinsic features. As such, the lack of adaptive immune responses in metastases with subtype 1 or 3 patterns may have important consequences concerning the responsiveness of such metastases to immune checkpoint therapies; however, the DNA repair abnormalities in subtype 1 metastases may induce hypersensitivity to DNA-damaging agents, including PARP inhibitors, whereas enrichment for angiogenic signatures may predict sensitivity to bevacizumab in subtype 3 metastases.

Our study sheds light on the identification of patients with potentially curable metastatic disease that might benefit from one or more focal treatments of limited metastases, such as those patients with low or intermediate integrated risks, versus those whose primary treatment modality should be systemic therapies with or without regional therapies, including those patients with high integrated risk classification. Further investigation is necessary to comprehensively characterize the molecular features of oligometastases and to what extent these concepts apply to patients with widespread disease or histological types other than colorectal cancer. While significant advances have been made in the understanding of widespread, lethal metastatic dissemination[32–36], our study is the first to investigate the molecular basis of potentially curable colorectal liver metastases. These results may provide a framework for future studies that pertain to the molecular classification of patients with limited metastatic disease from different primary sites and metastatic location, thereby providing a paradigm shift in the treatment of patients with metastatic disease.

## Methods

**Patients.** We analyzed samples from 134 adults with liver metastases from primary CRC of which 121 metastases from independent patients successfully underwent molecular analysis (Supplementary Fig. 1). The characteristics of these patients are described in Table 1, Supplementary Data 1, and the Supplementary Methods. We utilized a retrospective clinical cohort study design to identify patients who received uniform treatment for limited (defined as 1–5 lesions involving one or both hepatic lobes), resectable de novo CRC liver metastasis (CRCLM) at two collaborating institutions. We obtained appropriate approval from Institutional Review Boards at each respective cancer center.

**Analytic platforms.** We performed microRNA (miRNA) profiling for 116 samples using Affymetrix miRNA 4.0 Arrays. We performed whole genome RNA sequencing for 95 samples using Illumina TruSeq Stranded Total RNA Sequencing. In addition, we performed hybrid capture genomic sequencing of liver metastases and matched normal liver specimens from 59 patients using the OncoPlus panel[24]. All sequencing was conducted on Illumina HiSeq sequencers. We performed microsatellite instability (MSI) analysis on 89 samples using the Promega MSI 1.2 clinical assay according to FDA approved guidelines. Clinical data were frozen on April 30, 2016 and molecular data were frozen on June 26, 2016. Overall survival (OS), defined as the interval between hepatic resection and death from any cause or until censoring at the time the patient was last known to be alive, was chosen as the optimal primary endpoint. The complete list of datasets is provided in Supplementary Data 1.

**Statistical analysis.** The statistical analysis included Fisher's exact tests for associations of categorical variables when there were two categories or Chi-square tests when there were three categories. Kaplan–Meier and Cox proportional hazard analyses were used to examine the associations of molecular features with clinical outcomes. Multiple testing corrections were performed using the Benjamini-Hochberg method. All reported P-values are two-sided. A complete description of the methods is available in the Supplementary Information.

**Data availability.** RNA Sequencing data have been deposited at the European Genome-phenome Archive (EGA), which is hosted by the EBI and the CRG, under accession number EGAS00001002945.

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

## Acknowledgements

This work was supported in part by the Virginia and D.K. Ludwig Fund for Cancer Research, as well as generous gifts from The Foglia Foundation and Mr. and Mrs. Vincent Foglia (R.R.W.). The funding sources had no role in the study design, collection/analysis/interpretation of data, writing of the report, or decision to submit the paper for publication. The corresponding author had full access to all the data in the study and had final responsibility for the decision to submit for publication. The authors are also grateful to Dr. Philip Connell, Dr. Michael Spiotto and Dr. Ainhoa Arina for their careful reviews of the manuscript and MaryAnn Regner for her assistance in performing MSI analysis.

## Author contributions

S.P.P., N.N.K., M.T., M.C.P., S.H., and R.R.W. designed the study. S.P.P., N.N.K., A.U., S. C.W., S.G., N.J., K.M., L.X., C.W., M.E.S., S.K., P.P., K.K., K.P.W., M.T., M.C.P., and R.R. W. collected data. S.P.P., N.N.K., L.H., J.P., M.B., M.F., J.P.S., S.K., J.A., K.P.W., M.C.P., S. H., and R.R.W. contributed to data analysis. S.P.P., N.N.K., L.H., J.P., K.P.W., M.C.P., S. H., and R.R.W. interpreted data. S.P.P., N.N.K., L.H., J.P., M.B., M.F., K.P.W., and R.R. W. designed the figures. S.P.P., N.N.K., M.C.P., S.H., and R.R.W. wrote the manuscript. All authors approved the final manuscript prior to submission.

## Additional information

**Competing interests:** The authors declare no competing interests.

