## [Peer Review File · Nature Communications]

Reviewers' comments:

Reviewer #1 (Expertise: CRC metastasis, Remarks to the Author):

In this manuscript the Authors describe a new method for colorectal cancer patient stratification upon liver metastasis resection. This is a pioneering study that, using integrative genome-wide analysis, defines three molecular subtypes (SNF) of colorectal cancer liver metastases that associate with clinical outcome. Notably, these subtypes differ from consensus molecular subtypes (CMS) that are used for primary tumor classification thus highlighting the importance of molecular characterization of metastatic lesions. However, similarly to primary lesions the most favourable outcome is detected in the subtype that is characterized by immune activation and infiltration whereas stromal and angiogenic signatures associate with subtypes with worse prognosis. In addition, the Authors identify mutations that are exclusive for each SNF subtype thus opening the possibility of different treatment regimes that could be applied in liver metastatic CRC. The study is well designed, experiments are well performed and most of the relevant patient data is included in the manuscript. Ideally, this study should also include the data on molecular profile of primary tumors, which will enable better understanding of the similarities/differences between primary tumors and liver metastasis. Although this is not the case the study is sufficiently novel to be recommended for publishing, as it is one of the first manuscripts that is trying to decipher the molecular basis of different metastatic virulence.

Major comments:

1. How many metastatic lesions were analysed from the patients that developed more than one liver metastasis? If more than one liver metastatic lesion of the same patient was analysed did the authors find heterogeneity of these lesions?
2. How SNF subtyping applies to non-resectable liver metastasis?
3. Please define in all overall survival curves whether the time following surgery refers to the years following primary tumor resection or the years following liver metastasis resection?
4. Please include the data on whether surgical resection was R0 or R1 and how this associates with SNF types?

Minor comments:

1. Please avoid figure duplication (e.g. Supplementary Figure 16B and Figure 4B)

Reviewer #2 (Expertise: Cancer Genomics, Remarks to the Author):

The authors present an integrative analysis of a large cohort of colorectal liver metastases. The multi-dimensional molecular data the authors collected covered miRNA expression, mRNA expression, and DNA variations, enabling comprehensive characterization of potential subtypes of colorectal liver metastases. Analyses in conjunction with clinical information revealed three subgroups of colorectal liver metastases with distinct molecular properties and statistically different survival expectations. The analytical methods are elegant, the findings are interesting, and the manuscript is well written. However, before the acceptance for publication, I have several technical concerns/suggestions.

1. The authors tried to identify patient clusters based solely on miRNA or mRNA expression data. Although the identified clusters did not show significant associations with the survival data, prominent patterns were demonstrated. When the authors switched the clustering methods to SNF, three clusters with prognostic values emerged. My concern is how SNF clusters overlapped with those clusters solely identified upon miRNA/mRNA expression data. Because SNF applies a

fusion technique to integrate data sets, it can theoretically identify cluster structures complementarily defined by both of the two data sets. However, when the authors tried to construct the classifier for SNF clusters, they only used mRNA expression data. I expect the authors can further illustrate how both of the two data sets together defined the three SNF clusters. Methods beyond SNF may also be applied to the data sets to demonstrate the independence of the finding on the analytical method (i.e., SNF).

2. The authors identified that the SNF subtyping is a good prognostic index that is independent of Clinical Risk Scores (CRSs) through multivariate Cox proportional hazard analysis. But the details of the multivariate analysis were not stated with sufficient clearance. Upon my understanding, CRSs were transformed into a binary variable by setting a threshold 2. My concern is whether the independence will still hold when other cutoffs are applied or even no binary transformation is used. A related technical question is how SNF subtyping is coded in the multivariate analysis. Was it treated as one cardinal variable or three binary variables? If one cardinal variable was used, how was the cardinality determined for the three SNF clusters? Based on the data presented in Figure S15, I did a Chi-squared test to examine the associations between SNF subtyping and CRS-based clusters. The results suggest that their association is statistically significant ($p < 0.05$), which prompted me to raise the third question.

3. The authors can cluster patients into two or three subgroups based on CRSs and then analyze the molecular properties of each subgroup in a way similar to analyzing SNF clusters. Can obvious molecular patterns be observed for patients with different CRSs? Are the molecular patterns consistent or inconsistent with those of SNF clusters? I believe this type of analysis can not only verify the novelty of SNF subtyping but also can investigate the molecular evidence of different CRSs.

Minor points:

1. The section "Molecular Characterization of Intrinsic Liver Metastasis Subtypes" only focused on the mRNA-based gene signatures enriched in the different SNF subtypes. This makes me wonder whether that the miRNA aspect is not as important as author claimed.
2. The method for training single-sample CMS classifier is absent.
3. For Figure 2A, the source of the MSK cohorts should be described.

Our specific responses to the reviewers' comments are as follows:

Reviewer #1:

Major comments:

1. How many metastatic lesions were analyzed from the patients that developed more than one liver metastasis? If more than one liver metastatic lesion of the same patient was analyzed did the authors find heterogeneity of these lesions?

We designed our study to mirror clinical practice where typically only one metastatic site is biopsied to inform clinical management. Recent evidence from Kumar et al published in *Nature Medicine* 2016 (PMID: 26928463) supports our methodology. In this article, the authors analyzed whole exome sequencing, array comparative genomic hybridization (CGH), and RNA expression profiling of metastatic lesions from men with prostate cancer. They found significant molecular heterogeneity between individuals, but limited diversity comparing multiple metastases within a particular patient. As such, they concluded that evaluating a single metastasis provides sufficient molecular information regarding major oncogenic driver alterations in patients with multiple metastases. We believe these results are relevant to our study as well as to analyses of other histologic types of cancers associated with metastatic dissemination.

2. How SNF subtyping applies to non-resectable liver metastasis?

The authors are interested in investigating the utility of SNF subtyping in unresectable liver metastasis, although for the current manuscript we specifically studied the potential role for SNF subtypes in identifying subsets of liver metastasis with favorable clinical outcomes after surgical resection. Unfortunately, no publicly available data sets of unresectable colorectal liver metastasis exist which contain both mRNA and miRNA expression profiling from which we could perform SNF subtyping. However, inasmuch as primary tumor subtyping of liver metastases demonstrated similar distributions of CMS subtypes between resectable and non-resectable liver metastases (see Figure 2A), we hypothesize that non-resectable liver metastases will, at least in part, also be classifiable by our proposed SNF subtypes.

3. Please define in all overall survival curves whether the time following surgery refers to the years following primary tumor resection or the years following liver metastasis resection?

The “years following surgery” refers to the time following liver metastasis resection. The authors have clarified this point in the Methods and Materials section on page 13 of the main text.

4. Please include the data on whether surgical resection was R0 or R1 and how this associates with SNF types?

84% of patients underwent R0 resection while 16% underwent R1 resection. There was no association between surgical resection margin and SNF subtype ($P=0.42$, Chi-Squared test for proportion across three groups). This result was added to Supplementary Figure 6 (bottom row, second graph, page 23 of supplement) while the graph demonstrating the association between CRS and SNF subtypes was moved to Supplementary Figure 15 (page 32 of supplement).

Minor comments:

1. Please avoid figure duplication (e.g. Supplementary Figure 16B and Figure 4B).

The authors have removed the upper panel of Supplementary Figure 16B which is presented in the left panel of Figure 4B (page 33 of supplement). We thank Reviewer #1 for identifying this error.

Reviewer #2:

Major comments:

1. The authors tried to identify patient clusters based solely on miRNA or mRNA expression data. Although the identified clusters did not show significant associations with the survival data, prominent patterns were demonstrated. When the authors switched the clustering methods to SNF, three clusters with prognostic values emerged. My concern is how SNF clusters overlapped with those clusters solely identified upon miRNA/mRNA expression data. Because SNF applies a fusion technique to integrate data sets, it can theoretically identify cluster structures complementarily defined by both of the two data sets. However, when the authors tried to construct the classifier for SNF clusters, they only used mRNA expression data. I expect the authors can further illustrate how both of the two data sets together defined the three SNF clusters. Methods beyond SNF may also be applied to the data sets to demonstrate the independence of the finding on the analytical method (i.e., SNF).

Based on the reviewer's suggestion we examined the overlap between SNF clusters and those clusters identified by either miRNA or mRNA consensus clustering. We found that no single miRNA or mRNA consensus cluster completely comprised a respective SNF group, but rather each SNF cluster consisted of a non-random mixture of mRNA and miRNA consensus clusters (see revised Supplementary Figure 5 on page 22 and description of methodology on page 10 of supplement). This finding supported that SNF provided complementary information through integration of mRNA and miRNA data.

To explore whether an independent integrated analysis of mRNA and miRNA data also identifies prognostic subgroups, we performed a clustering of clusters (CoC) analysis using k=4 groups from both mRNA and miRNA consensus clustering. CoC analysis identified three integrated subgroups with statistically significant differences in 10-year overall survivals between groups 1 and 2 (50%) vs. group 3 (24%) (Log-rank P=0.032). These findings supported that integrated clusters, but not individual miRNA or mRNA clusters, determine the prognostic subgroups of colorectal liver metastasis.

Lastly, to examine whether SNF subtypes can be validated in independent clinical cohorts, we developed a signature which distinguished favorable SNF2 metastases from unfavorable SNF1 and SNF3 metastases. However, no publicly available clinical data sets of liver metastases contain both mRNA and miRNA expression data. Therefore, we specifically identified the mRNA features which optimally distinguished SNF2 from SNF1 and SNF3 metastases using Predictive Analysis of Microarrays (PAM). It is important to note that the signature-defining mRNAs in this analysis could not be identified without prior knowledge of the integrated SNF groups which underscores the importance of both mRNA and miRNA patterns in our analysis.

2. The authors identified that the SNF subtyping is a good prognostic index that is independent of Clinical Risk Scores (CRSs) through multivariate Cox proportional hazard analysis. But the details of the multivariate analysis were not stated with sufficient clearance. Upon my understanding, CRSs were transformed into a binary variable by setting a threshold 2. My concern is whether the independence will still hold when other cutoffs are applied or even no binary transformation is used. A related technical question is how SNF subtyping is coded in the multivariate analysis. Was it treated as one cardinal variable or three binary variables? If one cardinal variable was used, how was the cardinality determined for the three SNF clusters? Based on the data presented in Figure S15, I did a Chi-squared test to

examine the associations between SNF subtyping and CRS-based clusters. The results suggest that their association is statistically significant ($p < 0.05$), which prompted me to raise the third question.

Based on the reviewer's concern we tested various combinations of SNF as nominal variables (SNF1 vs. SNF2 vs. SNF3 or SNF2 vs. SNF1, 3) and CRS either as a continuous or nominal variable (CRS high [≥ 2] vs. CRS low [< 2]). We found no combination of SNF and CRS resulted in a significant multivariate interaction term based on Chi-squared test. Additional details regarding the multivariate Cox proportional hazard analysis of SNF and CRS are now included in the legend of Supplementary Figure 15 (page 32 of the supplement).

3. The authors can cluster patients into two or three subgroups based on CRSs and then analyze the molecular properties of each subgroup in a way similar to analyzing SNF clusters. Can obvious molecular patterns be observed for patients with different CRSs? Are the molecular patterns consistent or inconsistent with those of SNF clusters? I believe this type of analysis can not only verify the novelty of SNF subtyping but also can investigate the molecular evidence of different CRSs.

We performed differential expression analyses comparing mRNA and miRNA expression patterns between CRS high (≥ 2) and CRS low (< 2) metastases and found no statistically significant differences in either mRNA or miRNA expression patterns between groups after multiple testing corrections. Complementary analyses utilizing EGSEA demonstrated that CRS high metastases exhibit enrichment for specific metabolic pathways not identified in our SNF analysis, including pentose phosphate and glutamine synthesis pathways. These findings suggest that SNF subtyping contributes complementary information to CRS.

Minor points:

1. The section "Molecular Characterization of Intrinsic Liver Metastasis Subtypes" only focused on the mRNA-based gene signatures enriched in the different SNF subtypes. This makes me wonder whether that the miRNA aspect is not as important as author claimed.

Importantly, SNF subtypes were defined based on the integrated analysis of mRNA and miRNA patterns. As such, SNF subtypes can be defined by either mRNA- or miRNA-specific patterns. Unfortunately, we are not aware of any existing bioinformatics tools that allow both mRNA- and miRNA-based pathway analyses. In the "Molecular Characterization of Intrinsic Liver Metastasis Subtypes" section, we examined EGSEA pathways based only on the mRNA patterns that uniquely define each SNF group which is a well-accepted approach to detect gene-level pathways associated with specific molecular groups. In addition, pathway-based analyses of miRNA patterns strongly rely on annotation of the miRNA target genes rather than the miRNAs themselves. We believe our pathway analysis focusing on mRNA expression patterns is consistent with the methodology of many genome-wide cancer analyses and does not undermine the importance of miRNA patterns in the biology of SNF subtypes. Further work in our laboratory is investigating the miRNA-mRNA interaction networks which underlie SNF subtypes.

2. The method for training single-sample CMS classifier is absent.

The authors have clarified on page 10 Section VI of the Supplementary Methods and Materials section that the single sample method is a Pearson correlation-based centroid model of 786 genes which is included with the CMSClassifier R package. Training was not required for application of this centroid-based classifier.

3. For Figure 2A, the source of the MSK cohorts should be described.

Further description of the three MSK cohorts has been added to Section VI of the Supplementary Methods and Materials (page 9 of the supplement).

Thank you for your consideration of our work.

Reviewers' comments:

Reviewer #2 (Expertise: Cancer Genomics, Remarks to the Author):

The authors have addressed almost all my concerns satisfactorily except the independence of SNF typing and CRS grouping. In the rebuttal letter, the authors claimed "no combination of SNF and CRS resulted in a significant multivariate interaction term based on Chi-squared test." The authors were correct. But no interaction does not mean independence between two variables. In Supplementary Figure 15, the authors used Fisher's exact test to examine the dependence between SNF types and CRS groups and showed SNF type 1 is significant. According to my computation, SNF type 2 is significant, too. Here rxc chi-squared test is properer than Fisher's exact test to test the overall association between SNF typing and CRS-based grouping. Furthermore, I suggest the authors to rename the types because SNF is only a tool and cannot reveal the essence of the typing. The authors have also demonstrated that other integrative methods can result in similar observations. Overall, this manuscript is well written and provides important molecular details for CRC liver metastasis, deserving publication in Nature communications in my opinion.

Our specific responses to the reviewer's comments are as follows:

Reviewer #2:

1. The authors have addressed almost all my concerns satisfactorily except the independence of SNF typing and CRS grouping. In the rebuttal letter, the authors claimed "no combination of SNF and CRS resulted in a significant multivariate interaction term based on Chi-squared test." The authors were correct. But no interaction does not mean independence between two variables. In Supplementary Figure 15, the authors used Fisher's exact test to examine the dependence between SNF types and CRS groups and showed SNF type 1 is significant. According to my computation, SNF type 2 is significant, too. Here χ^2 test is properer than Fisher's exact test to test the overall association between SNF typing and CRS-based grouping.

We agree with Reviewer #2 that a non-significant interaction term in a multivariate Cox proportional hazard analysis does not indicate independence between two variables. To address the reviewer's concern, we further examined the association between SNF subtypes and CRS groups with the data presented in Supplementary Figure 15. Using a Chi-squared test comparing SNF1 vs. SNF2 vs. SNF3 by CRS groups, we found no significant difference across the three subtypes ($P=0.09$). However, we assessed whether specific SNF subtypes were associated with higher CRS using binary comparisons of each SNF subtype versus the two other SNF subtypes (upper four rows) or each SNF subtype versus another individual SNF subtype (bottom four rows). The data are presented in the following table:

Contrast	Chi-Sq P-value	Fisher's P-value
1 vs 2,3	0.037	0.063
2 vs 1,3	0.12	0.14
3 vs 1,2	0.57	0.65
Contrast	Chi-Sq P-value	Fisher's P-value
1 vs 2	0.034	0.047
1 vs 3	0.12	0.18
2 vs 3	0.49	0.59

These results suggest that while no significant association was identified across all three SNF subtypes and CRS, a significant association was detectable for SNF1 as compared to SNF2 such that patients with the SNF1 subtype were more likely to exhibit higher CRS than SNF2. Based on these findings, we agree with the reviewer's concern regarding our statement on the independence of SNF and CRS. Accordingly, we modified the second sentence of the "Integrated Risk Stratification" Section by removing the text highlighted in red below:

"Multivariate Cox proportional hazard analysis indicated the prognostic impact of SNF-based molecular subtypes was statistically ~~independent of but~~ complementary to CRS (**Supplementary Figure 15**)."

2. Furthermore, I suggest the authors to rename the types because SNF is only a tool and cannot reveal the essence of the typing. The authors have also demonstrated that other integrative methods can result in similar observations.

Based on the reviewer's suggestion, we have renamed the subtypes throughout the Main Text, Figures, Tables, and Supplement.

REVIEWERS' COMMENTS:

Reviewer #2 (Remarks to the Author):

The authors have completely addressed my concerns. I have no more comments.